# Whole-genome comparison of two same-genotype macrolide-resistant *Bordetella pertussis* isolates collected in Japan

**Kentaro Koide**[1]*, **Yumi Uchitani**[2], **Takahiro Yamaguchi**[3], **Nao Otsuka**[1], **Masataka Goto**[1], **Tsuyoshi Kenri**[1], **Kazunari Kamachi**[1]

**1** Department of Bacteriology II, National Institute of Infectious Diseases, Tokyo, Japan, **2** Division of Microbiology, Tokyo Metropolitan Institute of Public Health, Tokyo, Japan, **3** Division of Microbiology, Osaka Institute of Public Health, Osaka, Japan

* kkoide@niid.go.jp

**Data Availability Statement:** All relevant data are within the manuscript and its Supporting Information files.

## Abstract

The emergence of macrolide-resistant *Bordetella pertussis* (MRBP) is a significant problem because it reduces treatment options for pertussis and exacerbates the severity and spread of the disease. MRBP has been widely prevalent in mainland China since the 2010s and has been sporadically detected in other Asian countries. In Japan, two MRBP clinical strains were first isolated in Tokyo and Osaka between June and July 2018. The isolates BP616 in Osaka and BP625 in Tokyo harbored the same virulence-associated allelic genes (including *ptxP1*, *ptxA1*, *prn1*, *fim3A*, and *fhaB3*) and MT195 genotype and exhibited similar antimicrobial susceptibility profiles. However, despite their simultaneous occurrence, a distinguishable epidemiological link between these isolates could not be established. To gain further insight into the genetic relationship between these isolates in this study, we performed whole-genome analyses. Phylogenetic analysis based on genome-wide single-nucleotide polymorphisms revealed that the isolates belonged to one of the three clades of Chinese MRBP isolates, but there were 11 single-nucleotide polymorphism differences between BP616 and BP625. Genome structure analysis revealed two large inversions (202 and 523 kbp) and one small transposition (3.8 kbp) between the genomes. These findings indicate that the two Japanese MRBP isolates are closely related to Chinese MRBP isolates but are genomically distinct, suggesting that they were introduced into Japan from mainland China through different transmission routes.

## Introduction

*Bordetella pertussis* is the causative agent of pertussis, also known as whooping cough. This bacterium spreads rapidly through droplet transmission and can infect individuals of all ages. While pertussis generally has mild effects on adults, it can be fatal for newborns and unvaccinated infants, as they are susceptible to respiratory failure, respiratory arrest, and other life-threatening complications. Antibiotic treatment, prophylaxis, and pertussis vaccination play

**Funding:** This study was supported by grants from the JSPS KAKENHI grant numbers JP21K15441 (K. Ko.) and JP21fk0108139 (K. Ka.). The funders had no role in study design, data collection, and interpretation, or the decision to submit the work for publication.

**Competing interests:** The authors have declared that no competing interests exist.

important roles in controlling the spread of pertussis. Macrolides such as erythromycin and clarithromycin are recommended antimicrobial agents for the treatment of pertussis. However, macrolide-resistant *B. pertussis* (MRBP) has been detected in several countries [1]. MRBP was first identified in Arizona, USA, in 1994 [2]. Subsequently, *B. pertussis* isolates exhibiting reduced susceptibility to macrolides have been reported in several countries, albeit infrequently. During the 2010s, a notable emergence and proliferation of MRBP occurred throughout mainland China [3], and sporadic cases have been reported in other Asian countries [4, 5]. In Japan, MRBP was first isolated in 2018 [6].

Macrolide resistance in *B. pertussis* is associated with the A2047G mutation in the 23S rRNA gene [7, 8]. The homologous mutation in each of the three copies of the 23S rRNA gene changes in the antibiotic binding site, reducing the effectiveness of macrolides [9]. To date, no other mechanism underlying macrolide resistance in *B. pertussis* has been identified. Although standardized breakpoints for detecting macrolide resistance in *B. pertussis* have not been established, almost all isolates with the A2047G mutation exhibit minimal inhibitory concentrations (MICs) of >256 μg/mL when tested with erythromycin using the Etest method [10, 11]. Furthermore, macrolide resistance in *B. pertussis* is strongly associated with unique genotypes. Multi-locus variable-number tandem-repeat analysis (MLVA) revealed that three genotypes, MT55, MT104, and MT195, were commonly found in MRBP isolates but not in macrolide-susceptible *B. pertussis* (MSBP) isolates [7, 12]. MRBP isolates were further characterized based on the allelic profile of the pertussis toxin promoter region (*ptxP*) and virulence-associated genes (including *ptxA*, *prn*, *fim3*, and *fhaB*). In Chinese isolates, the presence of *ptxP1* and *fhaB3* alleles was found to be closely associated with macrolide resistance [10, 13]. Polymerase chain reaction (PCR)-based MLVA typing has shown that the Chinese MRBP strains may have spread to East and Southeast Asia [4].

Whole-genome analysis is the preferred method to study the phylogenetic relationships among *B. pertussis* isolates [14]. Chinese MRBP isolates are classified into three clades related to MLVA genotypes, namely MT55, MT104, and MT195, in a phylogenetic tree based on single-nucleotide polymorphisms (SNPs) detected using short-read sequencing. Because SNPs are relatively stable genetic markers, MRBP isolates can be differentiated based on the position and number of SNPs, even within the same clade [13]. Moreover, the advancement of long-read sequencing technology enabled the identification of genomic structural variations. Diversity at the genomic and structural levels has been observed in *B. pertussis* isolates during epidemics [15]. Currently, whole-genome analysis is the most widely used typing method in *B. pertussis* research.

In Japan, two MRBP isolates, namely BP616 and BP625, were initially collected during the summer of 2018 from Osaka and Tokyo, respectively. However, despite their simultaneous occurrence, a distinguishable epidemiological link between these isolates could not be established. An epidemiological association between these MRBP isolates could help comprehend the transmission pathways of MRBP in Japan. Although PCR-based typing and whole-genome analysis were performed for BP616 [6, 16], such analyses were not conducted for BP625. Therefore, in this study, we performed whole-genome sequencing of BP625 and aimed to investigate the genetic relationship between these two MRBP isolates based on SNP and genome structure analyses.

## Materials and methods

### MRBP isolates and antimicrobial susceptibility testing

The MRBP isolate BP625 was collected from a one-month-old boy in Tokyo on June 28th, 2018. BP616 was collected from a two-month-old boy in Osaka in July 2018, as described

previously (Table 1) [6]. The patient with BP625 presented with a severe cough and was administered intravenous azithromycin (10 mg/kg body weight per day) for five days, and the patient recovered after 17 days of hospitalization. The information was acquired on April 16$^{th}$, 2022, without accessing any details that could identify individual participants. Both isolates possessed the A2047G mutation in the 23S rRNA, which was confirmed using the Cycleave real-time PCR assay and PCR-based sequencing [17]. Antimicrobial susceptibility was assessed using the Etest method with a testing strip (bioMérieux, Craponne, France) following the manufacturer's guidelines. Briefly, a 0.5-McFarland standard equivalent inoculum was prepared and inoculated on Bordet-Gengou agar (Difco) containing 1% (w/v) glycerol and 15% (v/v) defibrinated horse blood. Minimal inhibitory concentrations (MICs) were determined for the following eight antibiotics: erythromycin, clarithromycin, piperacillin, ampicillin, trimethoprim/sulfamethoxazole, gentamicin, ciprofloxacin, and meropenem. The Etest strips saturated with these antibiotics were applied to the surface of the agar, and the MIC was determined by identifying the point at which the elliptical zone of inhibition intersected with the Etest strip after incubation at 36˚C for three to five days.

## PCR-based genotyping

PCR-based genotyping was performed using MLVA and multi-locus virulence-associated gene typing targeting *ptxP*, *ptxA*, *prn*, *fim3*, and *fhaB* alleles [18, 19]. The extracted DNA of BP625 was used as a template, and five variable-number tandem-repeat (VNTR) loci (VNTR1, VNTR3a/VNTR3b, VNTR4, VNTR5, and VNTR6) were amplified using PCR. The number of repeats at each VNTR locus was determined based on the DNA fragment length, and an MLVA type was assigned for each VNTR. The five virulence-associated allelic genes were amplified and sequenced. The allele type of each gene was determined by comparing the obtained sequences with those of designated alleles from data stored in the GenBank database (S1Table in S1 File).

## Whole-genome sequencing

The complete BP625 genome was sequenced using both short- and long-read sequencing methods. Genomic DNA was extracted using the Genomic-tip 100/G and DNA buffer set (Qiagen, Hilden, Germany). For short-read sequencing, a genomic library was prepared using the TruSeq DNA PCR-Free Kit (Illumina, San Diego, CA), followed by sequencing on the Illumina NovaSeq 6000 platform, generating 150-bp paired-end reads. Conversely, the long-read library was constructed using the SMRTbell Template Prep kit (Pacific Biosciences, Menlo Park, CA), and single-molecule real-time sequencing was performed using PacBio Sequel I (Pacific Biosciences, Menlo Park, CA). The Illumina platform yielded 6,411,695 reads, whereas the PacBio platform generated 176,974 subreads. All library preparation and sequencing steps were performed by Macrogen Co. (Tokyo, Japan). Before the downstream analysis, the raw

**Table 1. Information of MRBP isolates collected in Japan.**

| Isolate | Location (Prefecture) | Collection date | Patient's sex | Patient's age | Mutation in 23S rRNA gene | MLVA type | Virulence-associated allelic genes | Reference |
|---------|----------------------|-----------------|---------------|---------------|---------------------------|-----------|-----------------------------------|-----------|
| BP625 | Tokyo | June, 2018 | Male | 1 M | A2047G | MT195 | *ptxP1/ ptxA1/ prn1/ fim3A/ fhaB3* | This study |
| BP616 | Osaka | July, 2018 | Male | 2 M | A2047G | MT195 | *ptxP1/ ptxA1/ prn1/ fim3A/ fhaB3* | [6] |

M, month

read data were filtered using fastp version 0.20.1 for short reads (length of 20 bp; sequencing quality score of 20) [20] and Filtlong version 0.2.0 (https://github.com/rrwick/Filtlong) for long reads (length of 1,000 bp; 10% of poor-quality reads discarded).

## SNP extraction and phylogenetic typing

To identify SNPs in the BP625 genome, short reads were mapped against the complete genome sequence of BP616 (NCBI RefSeq accession number: NZ_AP024746.1) using the CLC Genomics Workbench version 20.2 (CLC Bio, Aarhus, Denmark). The SNP call was filtered with a cutoff of >30 reads covering the SNP site, and >50% of the reads supported the SNP. During this process, SNPs in genes encoding transposase proteins were excluded because of the presence of numerous transposable DNA elements, such as IS*481* and IS*1002*, in the *B. pertussis* genome [21]. The presence of these highly homologous regions could have resulted in mismapping.

The phylogenetic relationships between BP616 and BP625 were analyzed using 114 MRBP and MSBP isolates collected from Japan and China. These isolates were randomly selected from samples collected between 2014 and 2018 in previous studies (S2 Table in S1 File) [13, 14, 22, 23]. SNPs in each isolate were detected using the aforementioned method, and a total of 748 SNPs were identified (S3 Table in S1 File). Based on the SNP profiles, a phylogenetic tree was constructed using the maximum likelihood method, and bootstrap support was estimated by running 1,000 replications using the IQ-TREE software version 2.0.3 [24]. Phylogenetic tree annotation and visualization were performed using iTOL v6 (https://itol.embl.de/) [25]. All software programs were run using the default parameters unless specified otherwise.

## Hybrid *de-novo* genome assembly

The complete genome sequence was obtained using a hybrid assembly strategy that combined the long- and short-sequence reads. A consensus sequence was constructed from long reads using Trycycler ver. 0.5.3 [26], and base errors were fixed with short reads using Polypolish and Paired-cell Overlapping Loops of Cards with Authorization with multiple rounds until no further changes were observed [27, 28]. The resulting genome sequence was annotated using the DFAST pipeline [29] and registered in the international nucleotide sequence databases via the DNA Data Bank of Japan (DDBJ) Nucleotide Sequence Submission System (accession number AP026936.1). IS*481* and IS*1002* located within the genome were identified by performing a BLAST search (blastn) using a minimum coverage of 50% and a 98% identity threshold against sequences registered in the ISfinder database (accession number M22031 for IS*481* and Z54268.1 for IS*1002*). Common and unique regions were detected using the BLAST Atlas and Unique Genome analysis tools available on GView Server (https://server.gview.ca/).

## Comparing the genomic structure

To compare the genomic structure of BP625 and BP616, a whole-genome sequence alignment was performed using ProgressiveMauve with default settings [16]. The genome alignments of BP616 and BP625 were verified using conventional PCR. To confirm the order and orientation of the genomic content, four PCR primer sets were designed (S4 Table in S1 File). The 15-μL PCR mixture consisted of 0.3 μL of the DNA extract, 0.3 U of KOD-FX DNA polymerase (TOYOBO, Osaka, Japan), 0.32 mM of dNTP, and 0.14 μM of each primer set. The PCR started with an initial denaturation step of 2 min at 94°C, followed by a five-cycle touchdown protocol. Each cycle consisted of 10 s at 98°C, 30 s at 72–67°C, and 30 s at 68°C. Once the touchdown temperature was reached, the PCR was repeated for 25 cycles, each for 10 s at

98˚C, 30 s at 67˚C, and 30 s at 68˚C, with a final step of 2 min at 72˚C. The PCR products were analyzed using electrophoresis on a 1% agarose gel.

## Data availability

The complete genome sequence of BP625 has been deposited in the DDBJ/European Molecular Biology Laboratory/GenBank under the accession number AP026936.1. The short-read sequence of BP625 has been deposited in the DDBJ Sequence Read Archive under the accession number DRR445791.

## Results

### MICs, genotype, and allelic genes

BP625 exhibited an MIC of 256 μg/mL for erythromycin and clarithromycin, confirming its classification as an MRBP isolate (Table 2). The antimicrobial susceptibility profile of BP625 was similar to that of BP616 for all antimicrobial agents tested.

Furthermore, the two strains shared the same MLVA type (MT195) and virulence-associated allelic gene profile (*ptxP1/ptxA1/prn1/fim3A/fhaB3*; Table 1).

### SNP-based phylogenetic analysis

BP625 exhibited 11 SNP differences in the protein-coding sequence (CDS) regions when compared with BP616. In the phylogenetic analysis (Fig 1), all Chinese MRBP isolates except one (B17005) were classified into three subclades (I, II, and III) based on 748 SNPs (S5 Table in S1 File). BP616 and BP625 belonged to subclade I, corresponding to the MT195 genotype, *ptxP1* allele, and macrolide resistance. The Japanese MRBP isolates were most closely related to the Chinese MRBP isolates B14081 and B180980, with 6–7 SNP differences. Chinese MRBP isolates were collected in Beijing in different years (2014 and 2018) [14]; however, there were only two SNP differences between their genomes.

Our Japanese MRBP isolates had 6–17 SNP differences (mean = 9.4 SNPs) compared with 27 Chinese MRBP isolates belonging to subclade I (S4 Table in S1 File). Similarly, the Chinese MRBP isolates had 0–19 SNP differences among themselves (mean = 7.9 SNPs). The SNP differences between the Japanese and Chinese MRBP isolates were within the range observed for the Chinese MRBP isolates, indicating a genetic relationship between these two isolates. Conversely, when comparing BP616 and BP625 with Japanese MSBP, they exhibited 54–116 SNP

**Table 2. MIC of antimicrobials.**

| Antimicrobial agent | MIC (μg/mL) | |
|---|---|---|
| | **BP625** | **BP616*** |
| Erythromycin | >256 | >256 |
| Clarithromycin | >256 | >256 |
| Piperacillin | <0.016 | <0.016 |
| Ampicillin | 0.064 | 0.094 |
| Trimethoprim/sulfamethoxazole | 0.064 | 0.094 |
| Ciprofloxacin | 0.012 | 0.012 |
| Gentamicin | 0.19 | 0.19 |
| Meropenem | 0.047 | 0.047 |

MIC, minimal inhibitory concentration

*Reference [6]

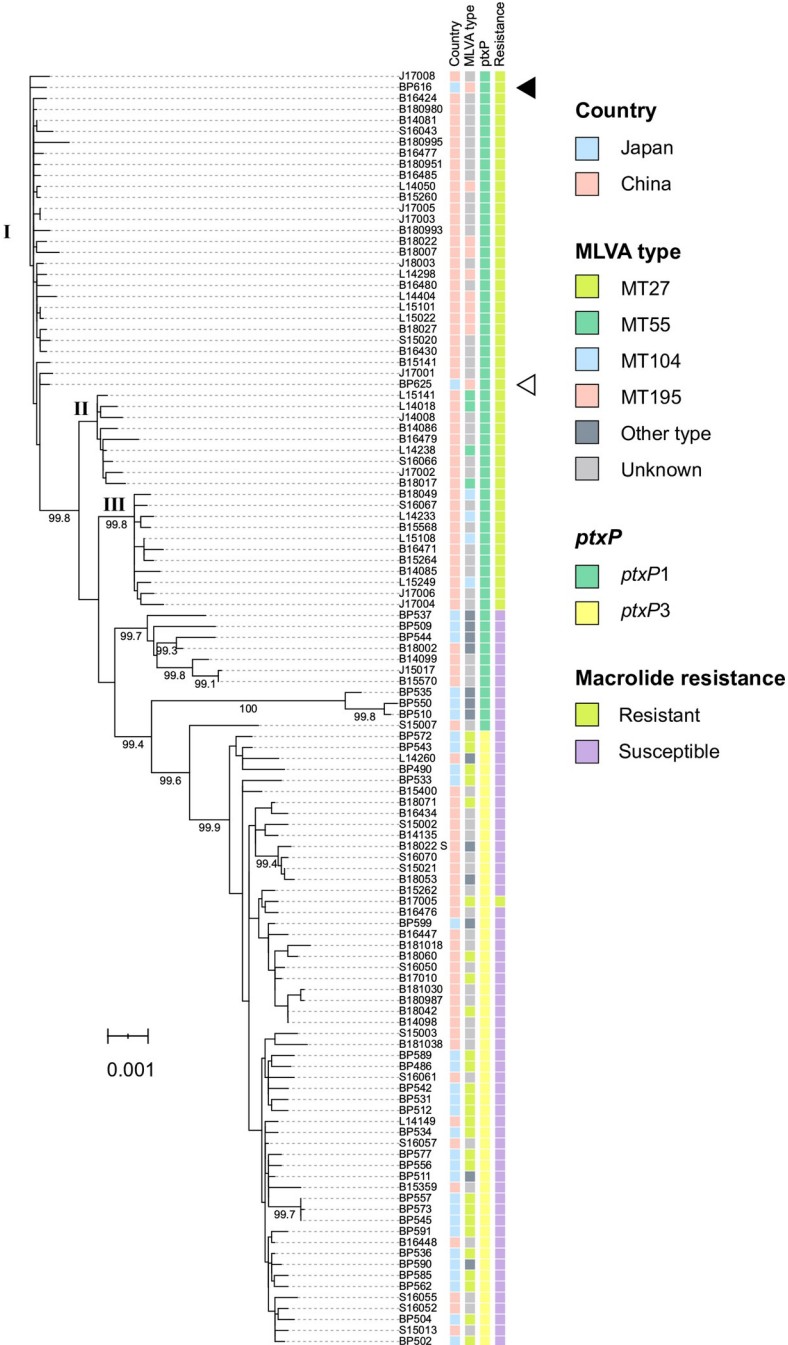

**Fig 1. Maximum likelihood phylogenetic tree for Japanese and Chinese *Bordetella pertussis* isolated between 2014 and 2018 based on single-nucleotide polymorphism.** BP616 was used as a reference genome and is included in the phylogenetic tree. Black and white arrowheads indicate the positions of BP616 and BP625, respectively. Bootstrap values greater than 98% are shown below the branches. Roman numerals I-III indicate the subclades of macrolide-resistant *B. pertussis*.

differences (S5 Table in S1 File). None of the Japanese MSBP isolates were closely related to the MRBP isolates BP616 and BP625.

## Genomic features of BP616 and BP625

The genome sequence of BP625 consisted of 4,132,330 base pairs (bp), which was 2,161 bp longer than that of BP616 (Table 3). Both genomes exhibited the same guanine + cytosine content (67.7%) and had six copies of IS*1002*. However, they had different numbers of genes (4,016 for BP616 and 3,984 for BP625) and IS*481* copies (252 for BP616 and 250 for BP625). The BLAST Atlas analysis showed no empty regions in the BP616 slot, indicating that most of the genes in the BP625 genome were also detected in BP616 (S1 Fig). However, Unique Genome analysis revealed a partial sequence of a gene encoding an ATP-binding protein that was unique to the BP625 genome. This gene was located between positions 2,018,251st bp and 2,019,600th bp in the BP625 genome, and the first 194 bp of this sequence showed no notable similarity in the BP616 genome.

## Comparison between genome structures of BP616 and BP625

The whole-genome alignment of BP616 and BP625 revealed six collinear blocks (Fig 2A). The BP625 genome exhibited two inversions, measuring 202 and 523 kb in length, and a transposition of 3.8 kb. PCR products corresponding to the P1 and P2 regions were detected in BP616 but not in BP625 (Fig 2B). In contrast, PCR products for the P3 and P4 regions were detected in BP625 but not in BP616. These PCR results confirmed the distinct genomic structures of BP616 and BP625.

## Discussion

In this study, we conducted whole-genome analysis of two MRBP isolates collected during the same period in Japan and identified distinct genetic differences between them. The isolates exhibited 11 SNP differences and variations in their genome structures. The data indicate the possibility of their introduction into Japan from China via different transmission routes. Additionally, the results indicated the potential of whole-genome analysis for determining strain identity in MRBP, which was difficult to determine using conventional PCR-based analysis such as MLVA typing.

Our MRBP isolates, BP616 and BP625, exhibited the same genotype and virulence-associated allelic profiles as determined via classical PCR-based typing. However, the whole-genome analysis revealed 11 SNP differences on CDS and variations in genome structures between the MRBP genomes. In a previous study, we proposed a threshold of three or less SNPs per genome for considering strains as the same in pertussis outbreaks [30]. According to this criterion, BP616 and BP625 were determined to be different strains. *B. pertussis* isolates from an outbreak typically share the same genomic arrangement [31]. However, the MRBP isolates exhibited different genomic structures (two inversions and one transposition) and a truncated gene encoding an ATP-binding protein that was unique to the BP625 genome. Based on these findings, we concluded that these MRBP isolates were different strains.

Our genome analysis strongly suggests that MRBP isolates BP616 and BP625 could have been introduced into Japan from China via different transmission routes. In the SNP-based phylogenetic tree, the isolates belonged to the same subclade (subclade I) as the Chinese MRBP isolates, indicating that they were closely associated with Chinese MRBP strains and were unrelated to the Japanese MSBP strains (Fig 1). The Chinese MRBP isolates B14081 and B180980, collected in Beijing in 2014 and 2018, respectively [14], were most closely related to the Japanese MRBP isolates. However, they exhibited 6–7 SNP differences compared with the Japanese MRBP isolates (S4 Table in S1 File). Considering these substantial SNP differences, it is unlikely that the Japanese MRBP isolates were directly introduced from Beijing to Japan. Some of the Chinese MRBP isolates from different areas, such as B16485 and B180951, showed

**Table 3. General genome features of Japanese MRBP.**

| Isolate | BP625 | BP616 |
|---|---|---|
| Size (base pairs) | 4,132,330 | 4,130,169 |
| G+C content (%) | 67.7 | 67.7 |
| Genes | 4,016 | 3,984 |
| Protein coding genes | 3,706 | 3,672 |
| Pseudo-genes | 246 | 248 |
| RNA | 64 | 64 |
| rRNA (5S, 16S, and 23S) | 3, 3, 3 | 3, 3, 3 |
| tRNA | 51 | 51 |
| ncRNAs | 4 | 4 |
| IS*481* | 252 | 250 |
| IS*1002* | 6 | 6 |
| NCBI accession | NZ_AP026936.1 | NZ_AP024746.1 |

less SNP differences compared with B14081 and B180980 than did BP616 and BP625. This suggested that MRBP may be spreading from the Chinese capital to other cities. Therefore, because they may have been brought in Japan via other Chinese cities, further studies involving more Chinese MRBP isolates from other cities are needed to verify this possibility.

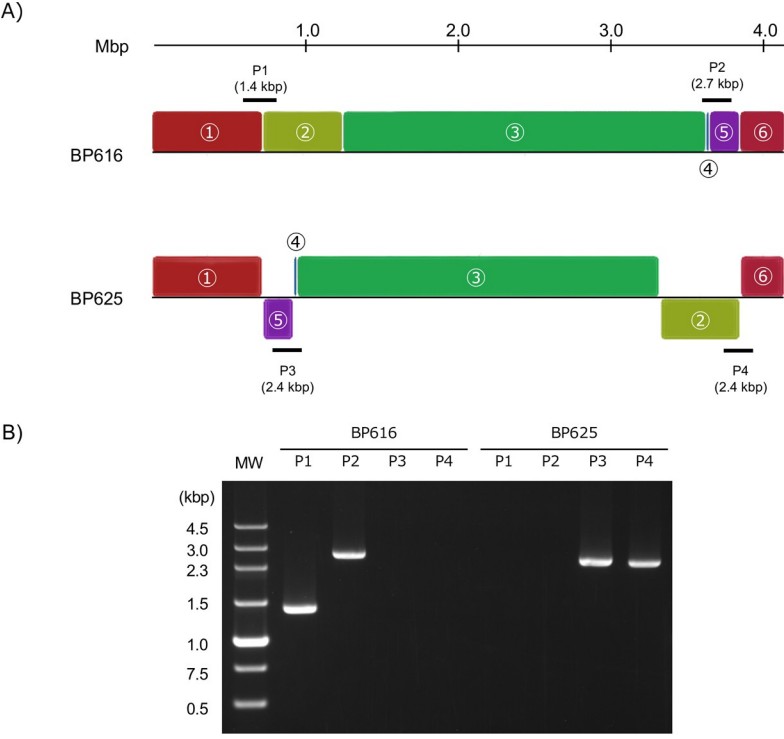

**Fig 2. Whole-genome alignment of BP616 and BP625.** A) Six locally collinear blocks were identified, which are shown as blocks of the same color. Inverted regions indicated by blocks of the same color but placed below the centerline were observed in the genomes of BP616 and BP625. Thick horizontal bars indicate target regions amplified via PCR to confirm the genomic structure, and the numbers in parentheses indicate the expected PCR product size. B) Agarose gel electrophoresis results of PCR using four primer sets. MW represents the molecular weight marker.

In epidemiological studies of infectious diseases, strain typing contributes to the understanding of bacterial transmission routes [32, 33]. In this study, MRBP isolates were effectively differentiated by identifying SNP differences through short-read sequencing. Out of 11 SNPs detected by mapping short reads of BP625 to BP616 genome, three SNPs were not identified in the CDS of the other isolates included in this study (S6 Table in S1 File). These unique mutations were not found in the sequences of any *B. pertussis* isolates other than BP625, as verified by the blastn program at the NCBI. The detection of the unique SNPs by short-read sequencing contribute to the elucidation of the transmission route of BP625 in Japan and China. In addition, short-read sequencing methods are generally more cost-effective, have higher throughput, and are more accurate than long-read sequencing technologies. Even if MRBP isolation increases in the future, short-read sequencing may be the best approach for investigating MRBP isolates in epidemiological studies.

In the late 2010s, a novel MRBP strain carrying the *ptxP3* allele emerged in mainland China [22, 34, 35]. *ptxP3* confers the strain with an increased pertussis toxin-production ability, and the *ptxP3* strain is considered to be more virulent in humans than the *ptxP1* strain according to a previous study on MSBP [36]. The shift from *ptxP1* to *ptxP3* has already occurred in MSBP, with the *ptxP3* strain being currently prevalent worldwide. This indicates that strains with *ptxP3* are more adaptable to the environment and spread more easily than those with *ptxP1*. *ptxP3*-MRBP isolates possess the MT27 or MT28 genotype, which are common genotypes among epidemic MSBP strains worldwide. A rapid shift from *ptxP1*- to *ptxP3*-MRBP strains has been observed in Shanghai, China, but the underlying cause remains unclear [34]. Moreover, the relationship between *ptxP1*- and *ptxP3*-MRBP strains, specifically if the *ptxP3*-MRBP strain emerges independently of *ptxP1*-MRBP, is unknown. Complete genome sequencing data enabled us to evaluate bacterial evolution based on the genome structure, providing insights into the relationships between MRBP strains. However, the availability of complete genome data for MRBP isolates is limited. Therefore, it is crucial to generate more publicly accessible complete sequence data to enable comparative analyses of the genome structures of MRBP isolates.

## Conclusion

The two Japanese MRBP isolates BP616 and BP625 may have been introduced from China via different transmission routes. Since the emergence and spread of MRBP in China, sporadic cases have been reported in East and Southeast Asia. Consequently, the need for epidemiological studies on MRBP is expected to increase in the coming years. Our findings indicate that whole-genome analysis, particularly short-read sequencing, is the most suitable method for epidemiological studies on MRBP.

## Supporting information

**S1 Fig. BLAST Atlas with BP625 as a reference genome.** The arrowhead indicates the region detected in the Unique Genome analysis.
(TIF)

**S1 File.**
(XLSX)

**S1 Raw images.**
(ZIP)

## Acknowledgments

We are grateful to T. Yamaoka (Toho University Omori Medical Center) and H. Kamiya (Immunization and Epidemiologic Research, National Institute of Infectious Diseases) for providing clinical information.

## Author Contributions

**Conceptualization:** Kentaro Koide, Kazunari Kamachi.

**Formal analysis:** Kentaro Koide.

**Investigation:** Kentaro Koide.

**Resources:** Yumi Uchitani, Takahiro Yamaguchi, Nao Otsuka, Tsuyoshi Kenri, Kazunari Kamachi.

**Writing – original draft:** Kentaro Koide, Kazunari Kamachi.

**Writing – review & editing:** Yumi Uchitani, Takahiro Yamaguchi, Nao Otsuka, Masataka Goto, Tsuyoshi Kenri, Kazunari Kamachi.

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
