## [Decision Letter · Decision Letter 0]

28 Nov 2023

PONE-D-23-27779Whole-genome comparison of two same-genotype macrolide-resistant Bordetella pertussis isolates collected in JapanPLOS ONE

Dear Dr. Koide,

Thank you for submitting your manuscript to PLOS ONE. After careful consideration, we feel that it has merit but does not fully meet PLOS ONE’s publication criteria as it currently stands. Therefore, we invite you to submit a revised version of the manuscript that addresses the points raised during the review process.

We look forward to receiving your revised manuscript.

Kind regards,

Daniela Flavia Hozbor

Academic Editor

PLOS ONE

Reviewers' comments:

Reviewer's Responses to Questions

**Comments to the Author**

1. Is the manuscript technically sound, and do the data support the conclusions?

Reviewer #1: Partly

Reviewer #2: Yes

Reviewer #3: Yes

2. Has the statistical analysis been performed appropriately and rigorously? 

Reviewer #1: N/A

Reviewer #2: N/A

Reviewer #3: Yes

3. Have the authors made all data underlying the findings in their manuscript fully available?

Reviewer #1: Yes

Reviewer #2: Yes

Reviewer #3: Yes

4. Is the manuscript presented in an intelligible fashion and written in standard English?

Reviewer #1: Yes

Reviewer #2: Yes

Reviewer #3: Yes

5. Review Comments to the Author

Reviewer #1: The manuscript submitted for publication in Plos One by Kentaro Koide et al entitled “Whole-genome comparison of two same-genotype macrolide-resistant Bordetella pertussis isolates collected in Japan” presents the genomic characterization of two macrolide-resistant Bordetella pertussis isolates collected in Tokyo and Osaka between June and July 2018.

One isolate was already reported in a previous publication (reference 5). This study provides 2 additional whole genome sequences of isolates resistant to macrolides recently collected in Japan and compares them to resistant isolates from China.

Abstract L32-34 should be removed: this is not the key message of this work.

L48- authors should mention that 3 copies of 23sRNA are present in the genome of B. pertussis and that resistant isolates mostly present the same mutation in the 3 copies of 23sRNA. Bartkus et al., 2003 publication (doi: 10.1128/JCM.41.3.1167-1172.2003) that first describes the mutation associated to erythromycin resistance in Bordetella should be added

L75-76: “However, despite their simultaneous occurrence, a distinguishable epidemiological link between these isolates could not be established => based on what?

L90-91: why where information acquired in April 2022 for isolates collected in 2018?

L91-93: where the 3 copies of 23sRNA mutated?

L105: why was PCR-based genotyping done since whole genome sequencing is performed?

L112-113: please give the accession Number of data from Genbank database for determination of alleles.

L124-L127. Information about reads quality should also be added.

L215: Figure 1: was the tree rooted? Which isolate correspond to the longer branch? It seems there is only 1 isolate ptxP3 and resistant in the dataset? Are MRBP from other country than China also included in the tree? Resolution of Figures 1&2 is not good on my pdf.

Authors used a SNP cut off determined in another study but were phylogenetic analysis the same as in the present study?

Authors should discuss mutations that differentiate the 2 MRBP isolates.

L293-304: this paragraph has to be shortened.

L470-475: Supporting information titles are not provided on the pdf.

Reviewer #2: Koide et al. investigated whole genome sequences of two macrolide-resistant Bordetella pertussis isolates in Japan. They show that 11 SNPs exist between two isolates, indicating these isolates independently transmitted from PRC to Japan. These information would contribute to understandings for Drug-resistant Bordetella pertussis.

Specific comments:

Line 186; Please correct 3256.

Line number should not add to Tables and foot note of Tables : Line 103-104, Line 191-202, Line 243-245.

Line 291: The authors described “They were likely introduced from other cities in China.” However, it is not shown the reason why Beijing, where the most similar strain was isolated, could be excluded from the origin.

Line 295: “Typically, short-read sequencing …”, This paragraph should discuss on what NGS technique is the best for analyzing MRBP genome in regard with transmission and subcluster. Descriptions in the manuscript is a bit too general.

Reviewer #3: Abstract:

- There is no previous sentence about the close relationship of your MRBP strains with Chinese strains. It is suggested to revise the sentence to clearly show that the phylogeny analysis suggested this close relationship.

- It is better to include a sentence in the abstract explaining why MRBP strains are important.

Introduction:

- Please rewrite lines 43-46 as a more accurate history of MRBP strains. The current sentence implies that MRBP strains were first found in China, which is not true. You also mentioned sporadic strains in Asian countries, but there are reports from other continents as well.

- Please rewrite lines 62-72 to make the paragraph more cohesive and clearer.

- The sentence in line 72 is ambiguous. Is WGS used widely in diagnostic labs or research labs? Please clarify this sentence.

- In line 131 why B616, one of the MRBP clinical strains, was chosen as a reference. Have you aligned your samples against TohamaI as well? The reviewer requested that you explain.

- What strain was used as an outgroup for phylogenetic analysis? Since most of the isolates were ptxP1, why did you not use the global reference genome Tohama I for SNP detection and phylogeny analysis?

- It is suggested to investigate the divergence time for MRBP isolates if possible.

- The authors did not explain and discuss the mechanisms of resistance to macrolides due to the mutation in 23srRNA either in the introduction or discussion.

-Since the authors emphasize the ptxP1 and ptxP3 strains in their manuscript it is suggested to explain their importance and genomic profiles either in the introduction or discussion section. For the majority of readers who do not have pertussis knowledge but are interested in antimicrobial resistance microorganisms, it is not clearly explained what ptxP1 and PtxP3 are and what is their role in BP and its virulence.

- It is suggested to the authors to discuss the importance of MRBP isolates. Since the majority of isolates in Japan are PtxP3 and rare ptxP1 isolates are isolated in current years why these strains are important.

- It is suggested to discuss why most of the MRBP isolates are ptxP1 and not ptxP3. Is there any ptxP profile information for MRBP strains of other countries?

6. PLOS authors have the option to publish the peer review history of their article (what does this mean?). If published, this will include your full peer review and any attached files.

Reviewer #1: No

Reviewer #2: No

Reviewer #3: No

---

## [Author Response · Author response to Decision Letter 0]

12 Jan 2024

Following are our point-by-point responses to the reviewers’ comments.

Comments from Reviewer 1

Comment 1: Abstract L32-34 should be removed: this is not the key message of this work.

Response: Thank you for your valuable feedback and suggestions. As recommended, we have removed the indicated sentences from the abstract.

Comment 2: L48- authors should mention that 3 copies of 23sRNA are present in the genome of B. pertussis and that resistant isolates mostly present the same mutation in the 3 copies of 23sRNA. Bartkus et al., 2003 publication (doi: 10.1128/JCM.41.3.1167-1172.2003) that first describes the mutation associated to erythromycin resistance in Bordetella should be added.

Response: We have added information on the history of MRBP (lines 44–49).

Comment 3: L75-76: “However, despite their simultaneous occurrence, a distinguishable epidemiological link between these isolates could not be established => based on what?

Response: Any possible common source of infection could not be found from the treating physicians’ comments. BP616 and BP625 were isolated from newborn patients who lived in two different prefectures, Osaka and Tokyo, respectively. The shortest distance between Tokyo and Osaka is approximately 400 kilometers, which makes it highly improbable that the two newborns were exposed to the same source of infection.

Comment 4: L90-91: why where information acquired in April 2022 for isolates collected in 2018?

Response: After the first report of MRBP in Japan in 2020, a Cycleave PCR assay was performed to detect the A2047G mutation in clinical isolates previously collected and stored at regional health laboratories in Japan. BP625 was found among isolates collected in Tokyo in 2018. Thus, the isolates collected in 2018 were only analyzed in 2022 after the first report in Japan in 2020.

Comment: L91-93: where the 3 copies of 23sRNA mutated?

Response: Yes, all three copies of 23S rRNA genes showed the A2047G mutation in Sanger and Cycleave PCR analyses. 

Comment: L105: why was PCR-based genotyping done since whole genome sequencing is performed?

Response: MLVA typing is a highly effective method for molecular epidemiology of B. pertussis. We performed MLVA typing of all isolates to infer the epidemiological links. In this study, as two MRBP isolates showed the same MLVA typing results, we were unable to determine the epidemiological relationship between the two isolates. Therefore, we performed whole genome analysis.

Comment: L112-113: please give the accession Number of data from Genbank database for determination of alleles.

Response: We have included a Supplemental Table listing the GenBank accession numbers.

Comment: L124-L127. Information about reads quality should also be added.

Response: We have included information about the quality of reads used in the analyses (lines 131-134).

Comment: L215: Figure 1: was the tree rooted? Which isolate correspond to the longer branch? It seems there is only 1 isolate ptxP3 and resistant in the dataset? Are MRBP from other country than China also included in the tree? Resolution of Figures 1&2 is not good on my pdf.

Response: Fig1 showed an unrooted phylogenetic tree including BP616 as a reference strain. Considering your queries, we have revised Fig 1 and 2. To make the tree more visually accessible, we revised the figure and adjusted the font size of the labels.. Additionally, we enhanced the resolution of the figures.

The ptxP3-MRBP is a newly found strain (Ref. 22 and 34) and, in this study, one MRBP isolate with ptxP3 was included. MRBP isolates collected in countries other than China and Japan were not included in the phylogenetic tree because reports of MRBP outside China are very rare and no sequence data are available for such isolates. 

Comment: Authors used a SNP cut off determined in another study but were phylogenetic analysis the same as in the present study?

Response: Our previous study indicated three or less SNPs in both CDS and intergenic regions as genetic markers of outbreak isolates. However, in this study, a difference of 11 SNPs in CDS was identified between BP616 and BP625. Therefore, the SNP difference between BP616 and BP625 would be larger when determined using the same method as in the previous study. To clarify the differences, we have rewritten the relevant sentences in the revised manuscript (lines 278–288).

Comment: Authors should discuss mutations that differentiate the 2 MRBP isolates.

Response: We have made S6 Table and added a sentence on SNP differences between BP616 and BP625 in the Discussion (lines 305–310). 

Comment: L293-304: this paragraph has to be shortened.

Response: We have revised this paragraph in accordance with the comments received from you and other reviewers (lines 304–315).

Comment: L470-475: Supporting information titles are not provided on the pdf.

Response: Thank you for pointing this out. We overlooked the need to include titles of the supplemental tables. We have included the same in the revised manuscript.

Comments from Reviewer 2

Comment: Line 186; Please correct 3256.

Response: We appreciate your thorough review of the manuscript. We have corrected the indicated section.

Comment: Line number should not add to Tables and foot note of Tables : Line 103-104, Line 191-202, Line 243-245.

Response: We appreciate your concerns on this point. However, according to the PLOS ONE manuscript body formatting guidelines, line numbers are to be included for tables and table footnotes. We have prepared the manuscript in accordance with these guidelines.

Comment: Line 291: The authors described “They were likely introduced from other cities in China.” However, it is not shown the reason why Beijing, where the most similar strain was isolated, could be excluded from the origin.

Response: We have rewritten the sentences to clarify the reason and our hypothesis (lines 298–303).

Comment: Line 295: “Typically, short-read sequencing …”, This paragraph should discuss on what NGS technique is the best for analyzing MRBP genome in regard with transmission and subcluster. Descriptions in the manuscript is a bit too general.

Response: We have rewritten and shortened this paragraph in accordance with your and other reviewers’ comments. (lines 304–315).

Comments from Reviewer 3

Comment: Abstract: There is no previous sentence about the close relationship of your MRBP strains with Chinese strains. It is suggested to revise the sentence to clearly show that the phylogeny analysis suggested this close relationship.

Response: We appreciate your comments regarding this point. We have added a sentence highlighting the fact that the phylogeny analysis suggested a close relationship of our MRBP strains with Chinese strains. 

Comment: It is better to include a sentence in the abstract explaining why MRBP strains are important.

Response: We have added a sentence at the beginning of the abstract to emphasize the importance of MRBP study.

Comment: Please rewrite lines 43-46 as a more accurate history of MRBP strains. The current sentence implies that MRBP strains were first found in China, which is not true. You also mentioned sporadic strains in Asian countries, but there are reports from other continents as well.

Response: We have provided a more detailed description of the historical context of MRBP after line 44.

Comment: Please rewrite lines 62-72 to make the paragraph more cohesive and clearer. The sentence in line 72 is ambiguous. Is WGS used widely in diagnostic labs or research labs? Please clarify this sentence.

Response: The applicability of WGS in diagnosis of pertussis has not yet been established. We have added references and rewritten the relevant text to emphasize that WGS is used in pertussis research (lines 67–76).

Comment: In line 131 why B616, one of the MRBP clinical strains, was chosen as a reference. Have you aligned your samples against TohamaI as well? The reviewer requested that you explain.

Response: The B. pertussis Tohama I strain was originally isolated in Japan in the 1950s. In this study, isolates collected between 2014 and 2018 were included in the phylogenetic analysis. The characteristics of the Tohama genome would be different from those of the tested isolates. For example, the genome size of BP616 is approximately 44 kbp longer than that of Tohama I (NCBI Accession: NC_002929.2). We surmised that using a reference genome sequence that is as closely related as possible would be beneficial for mapping short-read sequences. Therefore, we considered BP616 to be suitable as a reference sequence in the study on MRBP and have used the reference sequence in a previous study (Ref. 4).

In addition, we performed SNP identification and built an SNP-based phylogenetic tree using the Tohama I sequence. The overall shape of the phylogenetic tree was not different from that of the tree constructed using BP616 as the reference sequence; however, there were slight differences in the number and positions of SNPs among the isolates.

Comment: What strain was used as an outgroup for phylogenetic analysis? Since most of the isolates were ptxP1, why did you not use the global reference genome Tohama I for SNP detection and phylogeny analysis?

Response: We constructed an unrooted phylogenetic tree including BP616 as a reference strain. The objective of this study was to understand the relationship among Japanese and Chinese MRBP isolates that were recently collected. However, the B. pertussis Tohama I strain was a very old strain which was originally isolated in Japan in the 1950s. Rather than using the sequence of Tohama I, we used the sequence of BP616 as a reference for more accurate results.

Comment: It is suggested to investigate the divergence time for MRBP isolates if possible.

Response: Estimation of divergence time is not possible in this study because it requires data from strains collected over a long time period. We used isolates collected over a short period from 2014 to 2018. Thus, estimation of divergence time is a next step for future studies.

Comment: The authors did not explain and discuss the mechanisms of resistance to macrolides due to the mutation in 23srRNA either in the introduction or discussion.

Response: We have discussed the mechanism of macrolide resistance in the Introduction (lines 50–53).

Comment: Since the authors emphasize the ptxP1 and ptxP3 strains in their manuscript it is suggested to explain their importance and genomic profiles either in the introduction or discussion section. For the majority of readers who do not have pertussis knowledge but are interested in antimicrobial resistance microorganisms, it is not clearly explained what ptxP1 and PtxP3 are and what is their role in BP and its virulence.

Response: We have described the difference between ptxP1 and ptxP3 in the Discussion (lines 317–319).

Comment: It is suggested to the authors to discuss the importance of MRBP isolates. Since the majority of isolates in Japan are ptxP3 and rare ptxP1 isolates are isolated in current years why these strains are important.

Response: We have described the importance of MRBP and the difference between ptxP1 and ptxP3 in the Discussion (lines 317–322).

Comment: It is suggested to discuss why most of the MRBP isolates are ptxP1 and not ptxP3. Is there any ptxP profile information for MRBP strains of other countries?

Response: We appreciate your interest in additional information on the ptxP profile. However, studies on the predominance of ptxP1 among MRBP isolates in China as opposed to that of ptxP3 among MSBP isolates have been discussed in many other papers and have, therefore, not been included in this manuscript.

Because there are very few reports of MRBP strains isolated from countries other than China, information on ptxP is not available at all.

---

## [Editor Report · Decision Letter 1]

21 Jan 2024

Whole-genome comparison of two same-genotype macrolide-resistant Bordetella pertussis isolates collected in Japan

PONE-D-23-27779R1

Dear Dr. Kentaro Koide,

We’re pleased to inform you that your manuscript has been judged scientifically suitable for publication and will be formally accepted for publication once it meets all outstanding technical requirements.

Kind regards,

Daniela Flavia Hozbor

Academic Editor

PLOS ONE
---

## [Editor Report · Acceptance letter]

5 Feb 2024

PONE-D-23-27779R1 

PLOS ONE

Dear Dr. Koide, 

I'm pleased to inform you that your manuscript has been deemed suitable for publication in PLOS ONE. Congratulations! Your manuscript is now being handed over to our production team.

Kind regards, 

on behalf of

Dr. Daniela Flavia Hozbor 

Academic Editor

PLOS ONE